# Relationship between Bullying and the Type of Physical Activity Practised by Spanish Pre- and Adolescents

**DOI:** 10.3390/children10121888

**Published:** 2023-12-04

**Authors:** Juan de Dios Benítez-Sillero, Javier Murillo-Moraño, Diego Corredor-Corredor, Álvaro Morente-Montero, Luís Branquinho, José Manuel Armada-Crespo

**Affiliations:** 1Department of Specific Didactics, University of Cordoba, 14071 Cordoba, Spaineo1momoa@uco.es (Á.M.-M.); m62arcrj@uco.es (J.M.A.-C.); 2Laboratory of Studies on Coexistence and Violence Prevention (LAECOVI), 14071 Cordoba, Spain; 3Teacher Training College “Sagrado Corazón”, University of Cordoba, 14006 Cordoba, Spain; 4Counseling of Education, Junta de Andalucía, 14071 Cordoba, Spain; dcorcor015@g.educaand.es; 5Polytechnic Institute of Portalegre, Agrarian School of Elvas, 9000-082 Portalegre, Portugal; luis_branquinho@outlook.pt; 6Research Center in Sports Sciences, Health Sciences and Human Development, 6200-151 Covilhã, Portugal

**Keywords:** bullying, physical activity, adolescents

## Abstract

Background: The influence of bullying on physical activity beyond school time is uncertain, as it can vary widely in terms of type, modality, duration, adult supervision, and objectives. Methods: This study aims to analyze the relationship between school bullying and the type of physical activity practised. To this end, a descriptive study was made of 2025 pre- and adolescents aged between 10 and 19 years, reporting on their participation in victimisation and perpetration. The EBIPQ and PAQ-A were used. An analysis of the relationships between these variables was carried out according to gender and type of activity practised. Results: The results showed a higher rate of victimisation in boys who did not practise physical activity. Meanwhile, perpetration was higher in those who practised organised physical activity, especially in boys. Depending on the type of physical activity, the higher levels of both victimisation and perpetration of those who practised wrestling activities stand out in comparison with other groups. Conclusions: It could be stated that physical activity may be a protective factor against bullying victimisation, especially in boys. However, participation in organised physical activity activities may be related to higher perpetration in this sample in adolescent boys.

## 1. Introduction

The phenomenon known as bullying corresponds to a pattern of antisocial behaviour that results in a series of deliberate and negative behaviours based on an abuse of power by a specific student, or groups of students, over one or more of their peers, with the main characteristics of recidivism in the abusive behaviours and the intention to cause physical, social, and/or mental harm to the victim [1,2,3,4,5]. 

The victimised person suffers a series of problems associated with the bullying situation described as low self-esteem, emotional difficulties, shame, self-pity, social isolation, depression, anxiety, feelings of loneliness or suicidal ideation, among others [1,3,6]. Some studies present profiles that may be more sensitive to being bullied, such as women, students in secondary education, students with a low socioeconomic level, or adolescents outside the family context [1,7,8]. Physical activity contributes to the development of different dimensions of the personality [9] and has therefore become a context of interest to study the relationship with bullying. In most studies in relation to adolescence, it has been found that greater participation in physical activities, in addition to school Physical Education, is associated with less bullying victimisation [9,10,11,12,13,14,15], although other studies, to a lesser extent, have not found such a relationship [16,17,18,19]. Noteworthy is the study by Holbrook et al. [20], who reported lower victimisation in those who were more physically active but found no relationship with participation in sports-type activities. However, the relationship between physical activity and the role of the perpetrator or aggressor in bullying has been less well studied. In fact, students who participate in physical activity and sports may show less aggressive attitudes and more respectful behaviour towards their peers and rules and have greater self-control [21,22]. However, when we focus specifically on bullying perpetration behaviours, the results are scarce and not very enlightening. 

On the one hand, relationships have been found between greater participation in physical activity and greater bullying perpetration [18]. On the other hand, another study finds no relationship [16] and negative relationships between physical activity participation and cyberbullying in girls [23]. The variety of results found may be due to the lack of discrimination of specific aspects of the physical activity or sport practised, such as whether the physical activity is carried out freely, organised, competitive, or recreational, among other issues that determine its characteristics [23,24]. In this sense, the practice of organised physical activity does not guarantee inherent benefits, as positive experiences must be guaranteed in order to do so [21], although it does seem to bring other related benefits to adolescents who have been victims of bullying [14]. Likewise, it seems that the type of physical activity does have an influence on victimisation [18]. For example, participants in more physical contact sports activities show more aggressive behavioural tendencies than those who participate in non-contact or low-contact sports, recommending such activities, with a low level of participation, for bullying interventions [25]. Other authors highlight non-competitive physical activity as having the most positive results regarding a lower risk of developing aggressive behaviour [18], or that cooperative sports favour the reduction in bullying [22]. Medina Cascales and Prieto [26] found no differences in victimisation according to the type of physical activity and sport practised, which may be due to the small sample size.

In terms of gender, studies that address the issue in general show differences between boys and girls in terms of the type of bullying, with physical and verbal bullying being more prevalent in boys and relational bullying in girls [27,28]. In terms of physical activity as a differentiating factor, there is a greater aggressiveness and victimisation in boys who practice physical activity and a higher score in girls in terms of suffering from bullying [8]. Other studies point to a higher risk of developing aggressive or antisocial behaviours in boys and a lower risk of developing these behaviours in girls, in general, and the development of prosocial, respectful, and self-control behaviours in boys and girls who practise sports [21]. This issue contrasts with another study that concludes, based on the data collected in their research, that there are no correlations between physical activity and gender in bullying or victimisation [16]. In line with physical activity, gender, and bullying behaviours, the study carried out by [18] points to the participation of both males and females in bullying, finding trends towards a change in the levels of aggressiveness, more markedly so in boys than in girls, or less aggressive behaviour in students who participate in non-competitive team sports. The conclusions regarding victimisation in girls are noteworthy, as they showed less victimisation whether or not they took part in physical activity, which is consistent with another aspect highlighted in reference to the lack of influence of physical activity on levels of victimisation.

Another factor to consider in bullying behaviour in physical activity is age. Studies such as that of [21] confirm that age is a determining factor in the appearance and incidence of aggressive behaviour, an issue that is corroborated by other studies which point to the age of 11 to 14 years as the age that shows more aggressiveness in boys and girls, as well as being a phenomenon that tends to decline with the increase in the age of subjects [29]. Likewise, students aged 9–15 who participate in sports activities show behaviours with a lower risk of developing bullying, although some trends are observed in boys aged 13–15 regarding antisocial behaviour and neuroticism, both relevant variables in bullying [21].

In this sense, and considering the literature reviewed, it is observed that there are discrepancies in the description of victimisation and perpetration behaviours in bullying in reference to the type of physical activity practised, gender, and age, and that it is therefore necessary to know in detail the type of physical activity practised to better understand the relationships with bullying [8].

Therefore, we can hypothesise that adolescents involved in physical activity will be less involved in victimisation and less likely to be perpetrators of bullying, with further identification of such behaviours as a function of the type of physical activity practised, gender, and age.

The objective of this research is to examine the connections between victimisation and bullying behaviors, considering the variables of type of physical activity practised, gender, and age. The specific objectives we set ourselves are the following:-To analyse victimisation and perpetration behaviours according to the type of physical activity practised.-To analyse victimisation and perpetration behaviours as a function of physical activity as a function of gender and age.

## 2. Materials and Methods

### 2.1. Participants

In the present study, 2025 students participated in this study. Their ages ranged between 10 and 19 years, with a mean age (M) = 14.57, standard deviation (SD) = 1.74, and of which 1009 were girls (48.8%). The sample was selected by convenience based on accessibility, and participation of subjects was voluntary. They came from 7 public schools in southern Spain (6 in the province of Cordoba and 1 in the province of Huelva). The sample consisted of students from 5th year of Primary Education to 2nd year of Baccalaureate, in a medium socioeconomic context. 

### 2.2. Procedure

A descriptive, exploratory, cross-sectional study was carried out with non-probabilistic sampling. The present study was carried out after obtaining permission from the school councils of the participating schools, as well as duly signed and completed informed consent forms from the families. The inclusion criteria encompassed the willingness to participate and the completion of the consent form, while exclusion criteria involved the non-completion of the questionnaire. Participants were characterised by a moderate socio-economic level. This study protocol conformed to the latest version of the Declaration of Helsinki (2013), and the project was also approved by the Human Research Ethics Committee of the University of 11 December 2019. The average time for completing the questionnaire ranged between 20 and 30 min.

### 2.3. Instruments

#### 2.3.1. Bullying

The Spanish version of the European Bullying Intervention Project Questionnaire (EBIPQ) was used to measure the incidence of bullying [5]. It includes two dimensions addressing bullying victimisation and bullying perpetration. The first 7 questions are related to victimisation, and the last 7 to perpetration. It is composed of Likert-type response options, from 0 to 4, where 0 = never, 1 = once or twice, 2 = once or twice a month, 3 = about once a week, and 4 = more than once a week. Internal consistency values were equally optimal (overall victim α = 0.841 and overall perpetrator α = 0.805).

#### 2.3.2. Physical Activity

Two inquiries were posed to ascertain the quantity and nature of physical activity.

The initial question was derived from the primary query of the PAQ-A questionnaire [30] and underwent modification. The formulated questions were as follows:-Physical activity in your free time: Have you done any physical activity in the last 7 days (last week)? If yes, how many days?-Do you regularly attend any kind of physical activity classes, sports...? Indicate type of activity and days of the week.

Students answer with a number from 0 to 7 according to the number of days per week they practice.

From these two questions, the number of days of leisure time physical activity and participation in organised physical activities of each student were determined. Leisure-time physical activity was considered to include both free and organised practical activities, while organised activity was repetitive over time, dependent on a club or organisation, and led by an individual. In both cases, the compulsory days corresponding to the subject of Physical Education were not counted.

To analyze the nature of physical activity, the open-ended responses to question 2 were categorised into the following groups: non-practitioners, individual activities (athletics, cycling, and swimming), fitness (pilates classes, CrossFit, strength training, etc.), dance classes, rhythmic gymnastics, individual racket sports (tennis and badminton), paired racket sports (padel tennis), combat sports (karate, judo, kickboxing, and boxing), volleyball, team sports (basketball and handball), and football. In order to carry out this differentiation, the motor praxeology of Parlebas and previous studies were taken as a reference [23,31]. Volleyball, being a non-contact sport, was separated from other team sports. Football was also specifically analyzed due to its widespread popularity and distinctive social characteristics in our country [23]. 

### 2.4. Statistical Analysis

Data are presented as mean ± standard deviation (SD). The normality of the data distribution was assessed through the Kolmogorov–Smirnov test. All variables analysed had a non-normal distribution, so non-parametric techniques were applied. First, bivariate correlations were performed using Spearman’s test. Subsequently, comparisons between two independent groups were performed using the Mann–Whitney U-test. In addition, multiple linear regressions were performed on the dependent variables victimisation and perpetration. For the analysis of the types of physical activities, the Kruskal–Wallis test was performed for intergroup comparisons and the Mann–Whitney U-test for intragroup comparisons according to gender. The effect size was calculated according to Cohen [32]. Values above 0.8, between 0.8 and 0.5, between 0.5 and 0.2, and below 0.2 were considered large, moderate, small, and trivial, respectively. Data coding and analysis were performed using SPSS, version 26. Statistical significance was set at *p* < 0.05.

## 3. Results

Table 1 shows the correlation between organised activity, total activity, victimisation, and perpetration. Practised activity was lower with increasing age. High levels of significance were obtained between total activity and organised activity (0.579). Focusing on the variables concerning bullying, correlations were found between perpetration with organised activity (0.064) and perpetration with victimisation (0.556).

Regarding Table 2, the most remarkable aspect is that boys who do not practice physical activity suffer more victimisation (*p* = 0.04), finding no other significant aspect regarding gender and victimisation and/or perpetration in the total physical activity practised.

On the other hand, Table 3 shows that the total number of people who practice physical activity, in this case, organised physical activity, are more perpetrators than those who do not practise physical activity (*p* = 0.04). This value is maintained in boys (*p* ≤ 0.01) and not in their female counterparts (*p* = 0.24), indicating that this behaviour is more developed by boys.

Regarding age and gender, and after applying a linear regression based on total physical activity (Table 4), it was found that perpetration increases at higher ages (β = 0.057) and is significantly more frequent in boys (β = 0.105) compared to girls.

When repeating the linear regression model with the same independent variables, but in this case varying the dependent variable of organised physical activity (Table 5), we find that, as in Table 4, perpetration increases with increasing age (β = 0.059). In this case, there are significant relationships between organised physical activity and perpetration, indicating an increase in this behaviour in this activity (β = 0.043).

Table 6 examines the associations between the specific type of physical sports activity engaged in and experiences of victimisation or perpetration. Significant differences in victimisation were found to be higher among those who practise wrestling sports than among those who do not practise or carry out activities such as football, other team sports, and fitness training. In relation to perpetration, practitioners of wrestling activities presented significantly higher values than those who do not practice physical activity or carry out physical sports activities such as individual, dance, aquatics, rhythmic gymnastics, or other team sports not including football.

On the other hand, and analysing gender differences and the type of physical activity practised (Table 7), we found values indicating that boys are more aggressive than girls in racket sports (*p* = 0.05), wrestling (*p* = 0.00), football (*p* = 0.00), and fitness activities (*p* = 0.05).

## 4. Discussion

The purpose of this study was to analyse victimisation and perpetration behaviours in bullying among adolescents according to the type of physical activity practised, gender, and age.

As the main results of the study, we found that there is no relationship with total physical activity, i.e., including both organised and freely practised physical activity by adolescents. However, there is a difference between boys who engage in physical activity and those who do not, with less victimization observed in those who participate in physical activity. Meanwhile, perpetration is positively correlated with organised physical activity, showing differences in the group of boys, which implies that those who practise organised physical activity were more often perpetrators.

The data indicating that boys who are physically active experience less victimisation than boys who are not physically active are consistent with those of the majority of studies [9,10,11,12,13,14,15], although such relationships were not present in the sample as a whole, which is consistent with studies by different authors [16,17,18,19]. In this case, the results were not entirely conclusive, which could be due to the large number of characteristics that can differentiate the type of physical activity practised—for example, when we refer to quantity, frequency, company of other practitioners, organisation of the same by adults, or the type of physical sport activity in terms of its objectives or competitiveness among other factors [18,22,23,26,33].

Going into this detail at an initial level, we found that organised physical activity did not show any relationships or differences in victimisation behaviour but did show differences in perpetration, with boys being more likely to perpetrate the offence. It should also be noted that most studies do not distinguish between total, free, or organised physical activity. When comparing the existing literature, it should be pointed out firstly that perpetration is less studied than victimisation, and these data coincide with some studies [8,18], and no relationship was found in the study by Corral-Pernia [16]. This could corroborate the statement that the fact of practising organised physical activity does not guarantee a positive influence in this regard simply because of the fact of practising it [18,21]. The observed significance of differences in perpetration, particularly in relation to organised physical activity among boys, aligns with the reported findings of Méndez et al. [18], which found no differences in girls and no differences in boys and could be explained by the fact that boys tend to be more perpetrators than girls in relation to bullying [28] and specifically in physical activity [8]. 

In reference to age, no direct differences were found between bullying behaviours and bullying behaviours, which tends to contrast with different studies that find a decrease in bullying behaviours with age [29,34]. However, when physical activity and gender are included in the regression models, a greater perpetration is observed, coinciding with the trends found by Pelegrín Muñoz et al. [21].

Regarding the specific type of physical activity practised in relation to victimisation and perpetration, wrestling sports present significantly higher values in perpetration and victimisation compared to the rest of the activities practised and non-practitioners, which had been previously described [23,35]. This contrasts with experiences that contradict the findings, as there are studies that support the opposite in fighting sports such as judo, according to which they have a positive influence on the direct prevention of bullying and on variables related to it [36]. In this type of modality, in addition, practitioners have a high level of muscular strength in comparison with the practice of other physical sports activities [37], and greater muscle strength is related to greater perpetration of bullying in boys [38].

In terms of gender and physical activity, perpetration values are higher in boys, as in other studies [18,21,23,27], and in this case, in racket sports, wrestling, football and fitness modalities. Some possible explanations for these gender differences by sport modality could be due to the fact that girls tend to choose less competitive and contact-type activities in their sport physical activity practice than boys [21,23]. Likewise, girls who practice physical activity show greater empathy than their male peers, which could partially explain this relationship [33], and contact and competitive sports players have lower levels of empathy, especially boys.

## 5. Conclusions

From the results found, it can be concluded that the practice of physical activity may be a protective factor against bullying victimisation, especially in boys. However, participation in organised physical activity activities may be related to a higher perpetration, in this sample, in adolescent boys [18,28].

Within organised physical activities, participants in racket sports, wrestling, football, and fitness show higher levels of perpetration in boys than in girls. This could be related to the competitive nature, contact, or increased strength established in these activities.

This contrasts with the efficacy of sport–physical interventions in reducing aggressiveness [25,39,40,41] and bullying [17], including through combat sports such as judo [36] or martial arts [42], which, according to the data from our study, have been the groups with the worst results.

Therefore, it is proposed to develop an approach in the physical sports activities to be carried out by children and adolescents that takes into account the application of anti-bullying measures, improving the reduction in aggressiveness and improving aspects such as empathy or resilience from a specific approach that helps to build a more just and egalitarian society free of bullying, based on intervention programmes that have proven their scientific validity.

Likewise, in future research, it would be of interest to study whether adolescents who show higher levels of perpetration and victimisation maintain such behaviours in other contexts or to consider the point of view of coaches and family members. Another focus of future research would be the possibility of analysing the type of bullying that adolescents engage in or whether anti-bullying programmes improve perpetration and victimisation behaviours in different contexts of student interaction.

The main limitations of the study are based on the nature of self-reporting by the participants themselves, which could present some bias. Likewise, the sample is not randomised and does not represent the globality of a geographical territory.

## Figures and Tables

**Table 1 children-10-01888-t001:** Correlations between the different variables and victimisation and perpetration in bullying.

Variable	Age	Total Activity	Organised Activity	Victimisation
Total activity	−0.057 *	-	-	-
Organised activity	−0.058 **	0.579 **	-	-
Victimisation	−0.020	−0.022	−0.009	-
Perpetration	0.034	0.013	0.064 **	0.556 **

* Established level of significance; *p* < 0.05, ** Established level of significance; *p* < 0.01.

**Table 2 children-10-01888-t002:** Differences in the practice of physical activity in their free time between victimisation and perpetration (total physical activity).

	Total				Boys				Girls			
Condition	Mean ± SD				Mean ± SD				Mean ± SD			
	Yes (*n* = 1 602)	No (*n* = 423)	ES	*p*	Yes (*n* = 853)	No (*n* = 163)	ES	*p*	Yes (*n* = 749)	No (*n* = 260)	ES	*p*
Victimisation	0.43 ± 0.62	0.48 ± 0.63	0.08	0.10	0.41 ± 0.60	0.53 ± 0.70	0.01	0.04	0.45 ± 0.64	0.45 ± 0.58	0.06	0.96
Perpetration	0.23 ± 0.40	0.24 ± 0. 41	0.23	0.62	0.27 ± 0.46	0.27 ± 0.46	0.00	0.97	0.17 ± 0.32	0.21 ± 0.38	0.82	0.13

Notes. SD = standard deviation; ES = effect size. Established level of significance; *p* < 0.05.

**Table 3 children-10-01888-t003:** Differences in the practice of physical activity between victimisation and perpetration (organised physical activity).

	Total				Boys				Girls			
Condition	Mean ± SD				Mean ± SD				Mean ± SD			
	Yes (*n* = 1085)	No (*n* = 940)	ES	*p*	Yes (*n* = 613)	No (*n* = 403)	ES	*p*	Yes (*n* = 472)	No (*n* = 537)	ES	*p*
Victimisation	0.43 ± 0.60	0.45 ± 0.64	0.03	0.48	0.42 ± 0.60	0.44 ± 0.65	0.00	0.68	0.44 ± 0.62	0.46 ± 0.63	−0.06	0.67
Perpetration	0.25 ± 0.48	0.21 ± 0.39	0.23	0.04	0.30 ± 0.46	0.22 ± 0.42	0.18	<0.01	0.17 ± 0.30	0.20 ± 0.37	0.82	0.24

Notes. SD = standard deviation; ES = effect size. Established level of significance; *p* < 0.05.

**Table 4 children-10-01888-t004:** Linear regression of bullying on gender, age, and total physical activity.

Variable/Condition	Perpetrators		Victims	
	β	t	β	t
Gender female	−0.105 **	−4.677	0.018	0.819
Age	0.057 **	2.598	−0.040	−1.806
Total physical activity	0.012	0.532	−0.003	−0.128

Notes. β = Standarised Beta. ** Established level of significance; *p* < 0.01.

**Table 5 children-10-01888-t005:** Linear regression in relation to bullying determined by gender, age, and participation in organised physical activity.

Variable/Condition	Perpetration		Victimisation	
	β	t	β	t
Sex/Gender female	−0.098	−4.357	0.014	0.611
Age	0.059 **	2.670	−0.041	−1.859
Organised physical activity	0.043 **	1.917	−0.026	−1.139

Notes. β = Standarised Beta, ** Established level of significance; *p* < 0.01.

**Table 6 children-10-01888-t006:** Analysis of variations based on the type of physical activity undertaken in relation to experiences of victimisation or perpetration.

Total/Variable	Type P.A. (*n*)	Mean ± SD	Type P.A. (*n*)	Mean ± SD	ES	*p*
Victimisation	Fighting (107)	0.67 ± 0.78	Not practised (945)	0.45 ± 0.64	0.31	0.03
Victimisation	Fighting (107)	0.67 ± 0.78	Football (288)	0.38 ± 0.51	0.44	0.00
Victimisation	Fighting (107)	0.67 ± 0.78	Equipment (109)	0.34 ± 0.56	0.49	0.00
Victimisation	Fighting (107)	0.67 ± 0.78	Fitness (104)	0.36 ± 0.57	0.45	0.02
Perpetration	Fighting (107)	0.32 ± 0.56	Individuals (124)	0.17 ± 0.28	0.34	0.00
Perpetration	Fighting (107)	0.32 ± 0.56	Not practised (945)	0.18 ± 0.23	0.33	0.00
Perpetration	Fighting (107)	0.32 ± 0.56	Dance (143)	0.12 ± 0.23	0.47	0.00
Perpetration	Fighting (107)	0.32 ± 0.56	Aquatics (25)	0.03 ± 0.17	0.70	0.01
Perpetration	Fighting (107)	0.32 ± 0.56	Rhythmic (43)	0.05 ± 0.26	0.62	0.00
Perpetration	Fighting (107)	0.32 ± 0.56	Equipment (109)	0.13 ± 0.27	0.43	0.00

Notes. P.A. = physical activity; SD = standard deviation; ES = effect size. Established level of significance; *p* < 0.05.

**Table 7 children-10-01888-t007:** Analysis of the differences according to the type of physical activity practiced in the victimisation or perpetration (looking at differences between sex and type of F.A. performed).

Variable	Type P.A.	Boy (*n*)	Mean ± SD	Girl (*n*)	Mean ± SD	ES	*p*
Perpetration	Racket	31	0.30 ± 0.34	21	0.14 ± 0.20	0.52	0.05
Perpetration	Fight	75	0.53 ± 0.70	32	0.22 ± 0.24	0.59	0.00
Perpetration	Football	272	0.30 ± 0.46	16	0.11 ± 0.14	0.56	0.00
Perpetration	Fitness	54	0.34 ± 0.59	50	0.16 ± 0.24	0.40	0.05

Notes. P.A. = physical activity; SD = standard deviation; ES = effect size. Established level of significance; *p* < 0.05.

## Data Availability

The data presented in this study are available on request from the corresponding author. The data are not publicly available due to ethical and privacy considerations to protect the confidentiality of participants.

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
