# Peer review of "Relationship between Bullying and the Type of Physical Activity Practised by Spanish Pre- and Adolescents"

_children, 2023, doi:10.3390/children10121888_

Round 1

Reviewer 1 Report

Comments and Suggestions for Authors

Dear Authors,

It was interesting to read an article on a current topic.

The abstract and keywords of the article meet the requirements.

In the introductory part, the authors, while analyzing physical activity at a theoretical level, did not include all the research conducted in Europe (Lithuania, Portugal) on this topic in recent years (especially presenting the phenomenon of bullying in organized sports). 

I would think that the presentation and analysis of these studies would contribute to the quality of the article both in the introductory and discussion parts.

Also, the decision of the authors to categorize the activities of organized sports into certain types is not completely clear (see table 6). I would think that it would be appropriate to explain such a decision in the methodology part.

I would also suggest the authors to think about the need to cite other authors in the conclusions.

After making corrections in the mentioned areas, the article can be proposed for publication.

Reviewer 2 Report

Comments and Suggestions for Authors

 Dear Authors,

 Article appreciation: " Relationship between bullying and the type of physical activity  practised by Spanish pre- and adolescents"

 -The title fully corresponds to the study presented, in addition to presenting an adequate number of words.

 -The abstract presents the required elements, such as objectives, methodology, scope, sample, main results and conclusions.

-The introduction denotes conceptual elements necessary for the formulation of the problem, as well as evidencing the objectives of the study. Elements that show the relevance of the subject addressed are also presented. This paper presents a review of the current literature.

-The methodology is consistent with the study's proposal. Statistical analyses are appropriate to the proposed objectives.

-The discussion is adapted and integrates results and theoretical elements referred to in the text.

However, it is recommended that the authors include a reflection on the limitations of the research and the implications for future research.

It is suggested to change the sentence from line 118 to: “In the present study, 2025 students participated…”

The recommendation is: Accept with minor correction.

Round 2

Reviewer 1 Report

Comments and Suggestions for Authors

Dear authors, I hope my insights helped to improve the quality of the article

Author Response

Thank you very much for all the comments provided, as these clearly helped us improve the final version of the manuscript. We will proceed to make the suggested changes in green, as well as respond to each of your contributions.

R: The authors concluded  It could be stated that physical activity may be a protective factor against bullying victimization. Although this relationship is essential, it may not be that simple. However, seeing how authors controlled other confounding factors or variables may be interesting. Do children who participate in physical activity have other characteristics that may protect them from being victimized?
AUTHOR´S RESPONSE: Thank you very much for your suggestion. In the abstract, we were constrained by the journal's limit of 200 words, making it challenging to provide a detailed expansion of your question. However, in the conclusions section of the article, we specifically addressed your inquiry, offering a more comprehensive and detailed analysis.

R: 2. Materials and Methods-"A descriptive, exploratory, cross-sectional, descriptive study was carried out." The term descriptive is used twice in this sentence. Did the authors compare with another population without physical activity?
AUTHOR´S RESPONSE: Thank you very much. We have removed the repeated word, and regarding your question, our study was conducted in various schools, where students were surveyed about their participation in physical activities. Based on these responses, the corresponding analyses were carried out.

R: Discussion- Line 246: " main results of the study, we found that there is no relationship with total physical activity, i.e. that which includes organised physical activity plus that practised freely by
adolescents, but there is a difference between those who practise physical activity and
those who do not in boys, with less victimisation in those who practise physical activity."
This sentence is unclear and creates confusion about no relationship with physical activity.
I would suggest to perform a language revision.
AUTHOR´S RESPONSE: Thank you very much for your suggestion. We have modified that paragraph to make it more readable and better understood by the reader.